

# Identifying *MMP14* and *COL12A1* as a potential combination of prognostic biomarkers in pancreatic ductal adenocarcinoma using integrated bioinformatics analysis

Jingyi Ding[1], Yanxi Liu[2] and Yu Lai[3]

[1] Hospital of Chengdu University of Traditional Chinese Medicine, Chengdu, China
[2] University of California, Los Angeles, Los Angeles, CA, United States of America
[3] School of Basic Medicine, Chengdu University of Traditional Chinese Medicine, Chengdu, China

Corresponding author
Yu Lai, archimedean@rocketmail.com

## ABSTRACT

**Background**. Pancreatic ductal adenocarcinoma (PDAC) is a fatal malignant neoplasm. It is necessary to improve the understanding of the underlying molecular mechanisms and identify the key genes and signaling pathways involved in PDAC. **Methods**. The microarray datasets GSE28735, GSE62165, and GSE91035 were downloaded from the Gene Expression Omnibus. Differentially expressed genes (DEGs) were identified by integrated bioinformatics analysis, including protein–protein interaction (PPI) network, Gene Ontology (GO) enrichment, and Kyoto Encyclopedia of Genes and Genomes (KEGG) pathway enrichment analyses. The PPI network was established using the Search Tool for the Retrieval of Interacting Genes (STRING) and Cytoscape software. GO functional annotation and KEGG pathway analyses were performed using the Database for Annotation, Visualization, and Integrated Discovery. Hub genes were validated via the Gene Expression Profiling Interactive Analysis tool (GEPIA) and the Human Protein Atlas (HPA) website.
**Results**. A total of 263 DEGs (167 upregulated and 96 downregulated) were common to the three datasets. We used STRING and Cytoscape software to establish the PPI network and then identified key modules. From the PPI network, 225 nodes and 803 edges were selected. The most significant module, which comprised 11 DEGs, was identified using the Molecular Complex Detection plugin. The top 20 hub genes, which were filtered by the CytoHubba plugin, comprised *FN1*, *COL1A1*, *COL3A1*, *BGN*, *POSTN*, *FBN1*, *COL5A2*, *COL12A1*, *THBS2*, *COL6A3*, *VCAN*, *CDH11*, *MMP14*, *LTBP1*, *IGFBP5*, *ALB*, *CXCL12*, *FAP*, *MATN3*, and *COL8A1*. These genes were validated using The Cancer Genome Atlas (TCGA) and Genotype–Tissue Expression (GTEx) databases, and the encoded proteins were subsequently validated using the HPA website. The GO analysis results showed that the most significantly enriched biological process, cellular component, and molecular function terms among the 20 hub genes were cell adhesion, proteinaceous extracellular matrix, and calcium ion binding, respectively. The KEGG pathway analysis showed that the 20 hub genes were mainly enriched in ECM–receptor interaction, focal adhesion, PI3K-Akt signaling pathway, and protein digestion and absorption. These findings indicated that *FBN1* and *COL8A1* appear to be involved in the progression of PDAC. Moreover, patient survival analysis performed

via the GEPIA using TCGA and GTEx databases demonstrated that the expression levels of *COL12A1* and *MMP14* were correlated with a poor prognosis in PDAC patients ($p < 0.05$).

**Conclusions**. The results demonstrated that upregulation of *MMP14* and *COL12A1* is associated with poor overall survival, and these might be a combination of prognostic biomarkers in PDAC.

# INTRODUCTION

Pancreatic ductal adenocarcinoma (PDAC) is the most common malignant tumor of the pancreas and is a lethal malignancy with poor prognosis, which is in part due to its rapid progression and the lack of diagnostic and therapeutic targets. In 2018, pancreatic cancer (PC) ranked 11th among the most common cancers, with 458,918 new cases and 432,242 deaths due to PC worldwide (*Bray et al., 2018*). Recent work suggests that alcohol is a risk factor for PC (*Go, Gukovskaya & Pandol, 2005*), while both genetic and environmental factors also play a role in the development and progression of PC (*Piepoli et al., 2006*).

Understanding genetic alterations in the context of biological pathways can help identify specific novel biomarkers of PDAC. Previous studies identified several cancer-associated genes implicated in PDAC, including *KRAS* (*Waters & Der, 2018*), *MYC* (*Witkiewicz et al., 2015*), and *CDKN2A* (*Sikdar et al., 2018*). It is widely accepted that the formation of stroma contributes to tumor proliferation, invasion, and metastasis (*Von Ahrens et al., 2017*). Particularly pathognomonic for PDAC is a stromal reaction that occurs during tumor progression and extensively involves fibroblasts and the extracellular matrix (ECM) (*Mahadevan & Von Hoff, 2007*). Nevertheless, the precise etiology and pathogenetic mechanism of PDAC remain unclear.

Microarray technology provides high-throughput methods for quantitatively measuring the expression levels of thousands of genes simultaneously, and microarray-based gene expression profiling can filter differentially expressed genes (DEGs) and biological pathways linked to various malignant tumors. Therefore, microarray techniques are promising and efficient ways to identify candidate biomarkers involved in the pathogenesis of PDAC. The purpose of our study was to determine significant DEGs and pathways implicated in PDAC by integrated bioinformatics analysis and to provide novel insights into the progression, diagnosis, and therapeutic targets of PDAC.

# MATERIALS & METHODS

## Screening database

The Gene Expression Omnibus (GEO: https://www.ncbi.nlm.nih.gov/geo/) is a public repository of high-throughput gene expression genomics datasets (*Clough & Barrett, 2016*). In this study, we downloaded three microarray datasets, namely, GSE28735, GSE62165,

and GSE91035, from the NCBI-GEO database. The array data in GSE28735 consist of 45 matching pairs corresponding to PDAC and adjacent non-tumor tissues (*Zhang et al., 2013*; *Zhang et al., 2012*). GSE62165 includes data for 118 whole-tumor tissue and 13 control samples (*Janky et al., 2016*). GSE91035 incorporates data for 8 normal pancreatic and 25 PDAC tissues (*Sutaria et al., 2017*). Altogether, data for 188 PDAC tissues and 66 non-tumor tissues were available.

## Screening of DEGs

GEO2R (https://www.ncbi.nlm.nih.gov/geo/geo2r/) is an online analysis tool that is based on the R programming language and can be used to identify DEGs that differentiate between cancer and normal samples in a GEO series (*Yao & Liu, 2018*). Using GEO2R, we analyzed DEGs that differentiate between PDAC and non-tumor tissue samples. An adjusted $p$-value of <0.05 and |logFC| > 1 were employed as the cutoff criteria representing a significant difference. Using a data processing standard, we filtered DEGs via the Venn diagram tool at http://bioinformatics.psb.ugent.be/webtools/Venn/. A total of 263 DEGs were selected, which consisted of 167 upregulated genes and 96 downregulated genes.

## Establishment of the protein–protein interaction (PPI) network

The Search Tool for the Retrieval of Interacting Genes (STRING: http://string-db.org/) is an online application that can be used to assess DEG-encoded proteins and protein–protein interaction (PPI) networks (*Szklarczyk et al., 2015*). A combined score of >0.4 was set as the threshold.

Cytoscape software v3.2.1 (*Shannon et al., 2003*) was utilized to visualize the PPI network, which established a new way to find potential key candidate genes and core proteins. We utilized cluster analysis via the Molecular Complex Detection (MCODE) plugin with degree cutoff = 2, node score cutoff = 0.2, k-core = 2, and max depth = 100, which detected significant modules in the PPI network. To identify the hub genes, we also utilized the CytoHubba plugin, which provided a novel method of exploring significant nodes in PPI networks. These tools yield new insights into normal cellular processes, the underlying mechanisms of disease pathology, and clinical treatment.

## Gene Ontology (GO) and Kyoto Encyclopedia of Genes and Genomes (KEGG) pathway analysis of DEGs

The Gene Ontology (GO) is used to perform enrichment analysis, which covers the cellular component (CC), biological process (BP), and molecular function (MF), of the selected genes (*Young et al., 2010*). The Kyoto Encyclopedia of Genes and Genomes (KEGG) is a database that helps to illustrate the functionalities and pathways of the selected genes (*Altermann & Klaenhammer, 2005*). The Database for Annotation, Visualization, and Integrated Discovery (DAVID: http://david.ncifcrf.gov/) is a public online bioinformatics database (*Dennis Jr et al., 2003*) that contains information on biological functional annotations for genes and proteins. The cutoff criteria were selected on the basis of $p < 0.05$. We performed enrichment of the GO terms and KEGG pathways for the candidate DEGs using DAVID.

## Survival analysis of the candidate genes and validation of DEGs using TCGA and GTEx databases

Based on data for 9,736 tumors and 8,587 normal samples from The Cancer Genome Atlas (TCGA) database and the Genotype–Tissue Expression (GTEx) database, the Gene Expression Profiling Interactive Analysis tool (GEPIA: http://gepia.cancer-pku.cn/) is used to perform functions such as survival analysis, the detection of similar genes, and correlation analysis to clarify the relationships between diseases and DEGs (*Tang et al., 2017*).

The GEPIA was also utilized for validating and visualizing the selected DEGs using TCGA and GTEx databases (*Tang et al., 2017*).

## Validation of expression of candidate gene-encoded proteins

The expression of proteins encoded by the PDAC candidate genes was validated using the Human Protein Atlas (HPA: https://www.proteinatlas.org/) website on the basis of spatial proteomics data and quantitative transcriptomics data (RNA-Seq) obtained from immunohistochemical analysis of tissue microarrays.

## RESULTS

### Identification of DEGs

A total of 263 DEGs were identified from GSE28735, GSE62165, and GSE91035. There were 167 upregulated genes and 96 downregulated genes in PDAC tissues in comparison with non-tumor tissues (Fig. 1) (Table 1).

### Establishment of the PPI network

Using the STRING application and Cytoscape software, 225 nodes and 803 edges were mapped in the PPI network (Fig. 2A). In association with these nodes, the whole PPI network was analyzed using the MCODE plugin, and one significant module was identified with average MCODE score = 8.6, nodes = 11, and edges = 43 (Fig. 2B). This significant module comprised 11 DEGs, namely, *COL6A3*, *COL3A1*, *VCAN*, *COL5A2*, *COL12A1*, *THBS2*, *FBN1*, *POSTN*, *LTBP1*, *MMP14*, and *CDH11*. From the PPI network, the top 20 hub genes were filtered by the CytoHubba plugin using the maximal clique centrality method. Their order of sequence was as follows: *FN1*, *COL1A1*, *COL3A1*, *BGN*, *POSTN*, *FBN1*, *COL5A2*, *COL12A1*, *THBS2*, *COL6A3*, *VCAN*, *CDH11*, *MMP14*, *LTBP1*, *IGFBP5*, *ALB*, *CXCL12*, *FAP*, *MATN3*, and *COL8A1* (Fig. 2C). Via data mining, we found that the significant module and hub genes mainly consisted of upregulated genes.

### GO and KEGG pathway analysis of DEGs

Functional and pathway enrichment analyses were accomplished using DAVID. GO analysis showed that the most significant module was mainly enriched in cell adhesion, extracellular matrix structural constituent, and proteinaceous extracellular matrix (Fig. 3) (Table 2). Moreover, the 20 hub genes were mainly enriched in cell adhesion, endodermal cell differentiation, proteinaceous extracellular matrix, and calcium ion binding (Fig. 4) (Table 3). In addition, KEGG pathway enrichment analysis demonstrated that the DEGs in the most significant module were enriched in ECM–receptor interaction (Fig. 3) (Table 2)
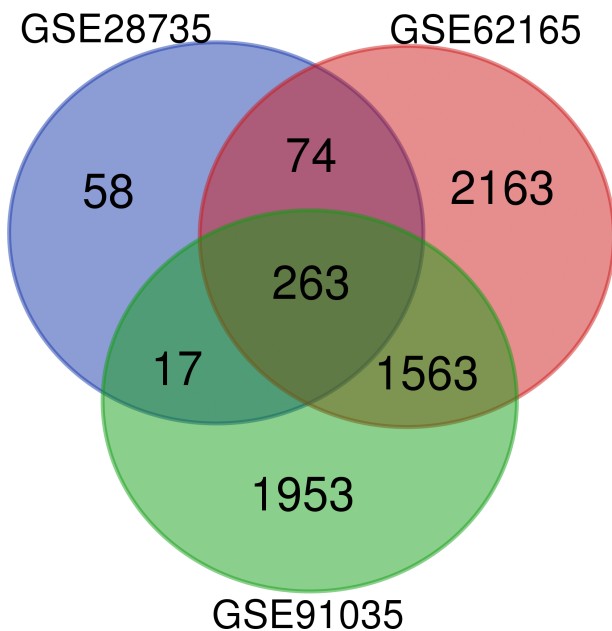

**Figure 1** **Venn diagram.** Identification of differentially expressed genes (DEGs) from GSE28735, GSE62165, and GSE91035. The different colored areas represent the different datasets, and a total of 263 DEGs were common to all three datasets.

**Table 1** **A total of 263 DEGs were identified from the three microarray datasets, which consisted of 167 upregulated genes and 96 downregulated genes present in pancreatic ductal adenocarcinoma (PDAC) tissues in comparison with non-tumor tissues.**

| DEGs | Gene names |
| --- | --- |
| Upregulated | *XDH, RTKN2, PTPRR, ADAM12, STYK1, TPX2, PADI1, HEPH, CEACAM6, ITGA3, COL1A1, ANLN, FNDC1, PCDH7, SLC6A6, TRIM29, PXDN, EDNRA, LTBP1, MFAP5, PLA2R1, FN1, KRT17, PGM2L1, IFI27, ASAP2, LAMB3, TNFAIP6, HOXB5, OAS1, NTM, COL5A2, OSBPL3, TMPRSS4, ANTXR1, SDR16C5, OLR1, NT5E, CTSK, SULF2, MXRA5, APOL1, CDH11, AREG, MALL, S100A16, BGN, LAMA3, COL8A1, IGFBP5, MMP12, ADAMTS6, SLC2A1, CD109, ECT2, KIF23, MMP11, CDH3, LMO7, CCL18, ATP2C2, POSTN, MMP14, ADAM28, SRPX2, CEACAM5, TMC5, OAS2, MUC17, GABRP, COMP, SYTL2, GPX8, RUNX2, DLGAP5, KRT19, VCAN, MKI67, SULF1, LAMC2, GCNT3, NMU, MUC13, CEACAM1, ETV1, COL12A1, AGR2, ST6GALNAC1, SLC44A4, PLAU, S100P, SERPINB5, FOXQ1, TGM2, ITGB4, DCBLD2, TRIM31, RAI14, NRP2, SGIP1, CST1, ARNTL2, LEF1, MYOF, ANO1, S100A14, DDX60, KYNU, CAPG, CCL20, MATN3, NPR3, GPRC5A, NOX4, IL1RAP, ACSL5, HPGD, GREM1, SCEL, FBN1, IGFL2, SLC6A14, KRT6A, DHRS9, ANGPT2, MST1R, COL3A1, TMEM45B, EDIL3, ASPM, FAP, INPP4B, LOXL2, NQO1, CYP2C18, IFI44L, HK2, EFNB2, AEBP1, SLC16A3, CORIN, THBS2, BCAS1, DSG3, DKK1, RHBDL2, COL17A1, TSPAN1, FERMT1, CXCL5, COL6A3, COL10A1, ACTA2, PLAC8, AHNAK2, MLPH, FBXO32, TGFBI, KCNN4, CLDN18, FGD6, MTMR11, FXYD3, MBOAT2, SEMA3C, DPYSL3, CENPF* |
| Downregulated | *EPB41L4B, GSTA2, KIAA1324, CELA3A, ACADL, CEL, SLC39A5, LONRF2, SLC3A1, NRG4, MT1G, PROX1, G6PC2, C5, EGF, FAM3B, AQP8, CLPS, SLC17A4, CPB1, GP2, PDK4, RBPJL, PDIA2, PM20D1, CTRC, IAPP, PLA2G1B, ERP27, CELA2B, GRP, REG1A, KIF1A, GUCA1C, CTRL, SYCN, CHRM3, TMED6, ALB, KCNJ16, REG3A, SLC4A4, AOX1, SERPINA5, CELA2A, SPINK1, FAM129A, FAM150B, SLC16A12, F11, CPA2, SV2B, BNIP3, C2CD4B, SLC1A2, REG1B, SCGN, PAK3, PRSS3, GRB14, REG3G, DCDC2, F8, GPHA2, EPHX2, PNLIPRP2, SLC7A2, CPA1, PRKAR2B, ONECUT1, BACE1, NUCB2, HOMER2, CXCL12, SLC43A1, GNMT, NR5A2, ALDH1A1, IL22RA1, BEX1, ANPEP, CFTR, FLRT2, LMO3, FGL1, NRCAM, FABP4, PNLIPRP1, KLK1, SERPINI2, GATM, DPP10, C6, SLC16A10, PRSS1, PAH* |
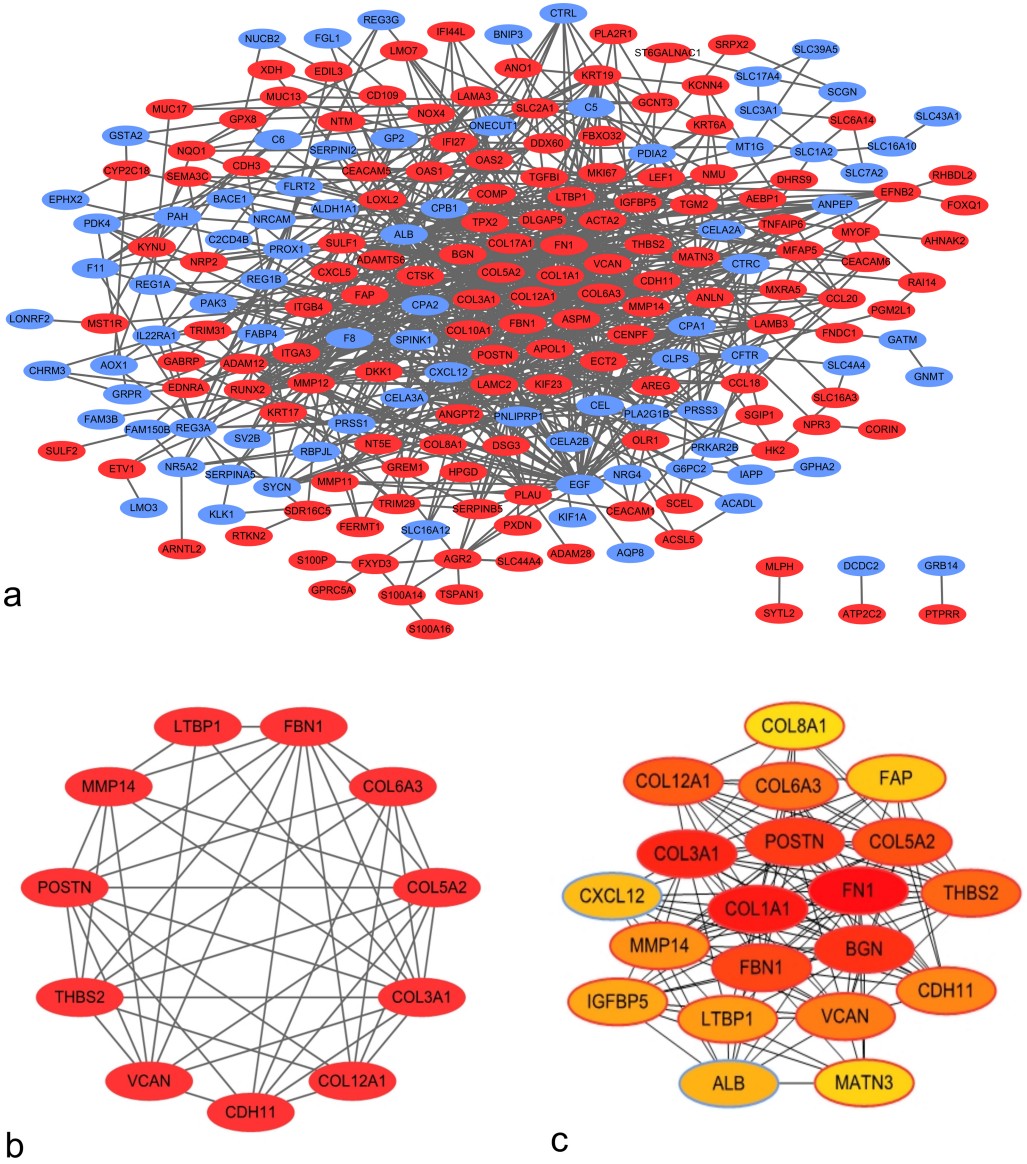

**Figure 2** **Protein–protein interaction (PPI) network of DEGs.** (A) PPI network of 263 DEGs in PDAC tissues. Red nodes represent upregulated genes, whereas blue nodes represent downregulated genes. (B) Significant module identified from PPI network via the Molecular Complex Detection plugin. This module consisted of upregulated genes. (C) Top 20 hub genes filtered using CytoHubba plugin. Nodes shown in darker colors were found to have higher significance. Red represents the highest significance, followed by orange, whereas yellow represents the lowest significance.

and the hub genes were mainly enriched in ECM–receptor interaction, focal adhesion, protein digestion and absorption, and PI3K-Akt signaling pathway (Fig. 4) (Table 3). (If $p < 0.0001$, the corresponding term was considered to be enriched.)

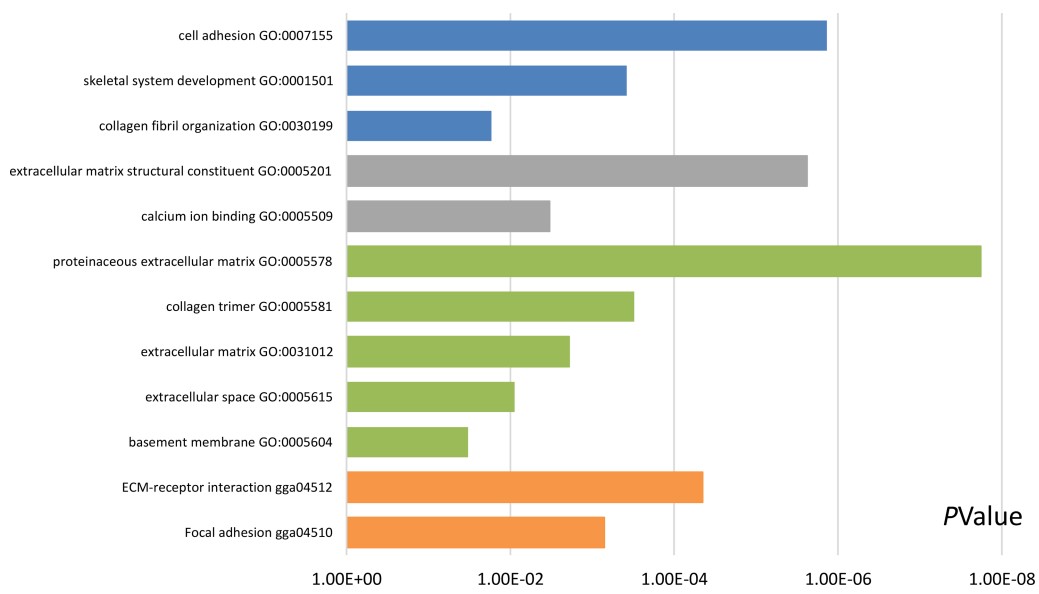

**Figure 3  Results of Gene Ontology (GO) and Kyoto Encyclopedia of Genes and Genomes (KEGG) pathway analyses of the most significant module.** The blue color represents biological process (BP), the gray color represents molecular function (MF), the green color represents cellular component (CC), and the orange color represents KEGG pathways.

**Table 2  Results of Gene Ontology (GO) and Kyoto Encyclopedia of Genes and Genomes (KEGG) pathway analyses of the most significant module.**

| Pathway ID | Pathway description | Count in gene set | *p*-value | FDR | DEGs |
|---|---|---|---|---|---|
| GO:0007155 | Cell adhesion | 5 | 1.37E−06 | 0.001121514 | COL6A3, COL12A1, VCAN, POSTN, THBS2 |
| GO:0001501 | Skeletal system development | 3 | 3.80E−04 | 0.310820683 | FBN1, VCAN, COL5A2 |
| GO:0030199 | Collagen fibril organization | 2 | 0.017003019 | 13.1195401 | COL5A2, CDH11 |
| GO:0005201 | Extracellular matrix structural constituent | 4 | 2.39E−06 | 0.001487027 | COL3A1, FBN1, VCAN, COL5A2 |
| GO:0005509 | Calcium ion binding | 4 | 0.003312243 | 2.045566581 | FBN1, VCAN, THBS2, CDH11 |
| GO:0005578 | Proteinaceous extracellular matrix | 6 | 1.81E−08 | 1.29E−05 | COL3A1, COL6A3, FBN1, COL12A1, VCAN, POSTN |
| GO:0005581 | Collagen trimer | 3 | 3.07E−04 | 0.217600344 | COL3A1, COL6A3, COL12A1 |
| GO:0031012 | Extracellular matrix | 3 | 0.001912518 | 1.350246354 | FBN1, COL12A1, THBS2 |
| GO:0005615 | Extracellular space | 4 | 0.008881226 | 6.138544445 | COL6A3, FBN1, COL12A1, POSTN |
| GO:0005604 | Basement membrane | 2 | 0.032936583 | 21.16653102 | FBN1, THBS2 |
| gga04512 | ECM–receptor interaction | 4 | 4.51E−05 | 0.019602271 | COL3A1, COL6A3, THBS2, COL5A2 |
| gga04510 | Focal adhesion | 4 | 7.12E−04 | 0.309146908 | COL3A1, COL6A3, THBS2, COL5A2 |

**Notes.**
Abbreviation: FDR, false discovery rate.

## Overall survival analysis of the top 20 hub genes

Patient survival analysis performed via the GEPIA using TCGA and GTEx databases demonstrated that the high expression levels of *COL12A1* and *MMP14* were correlated

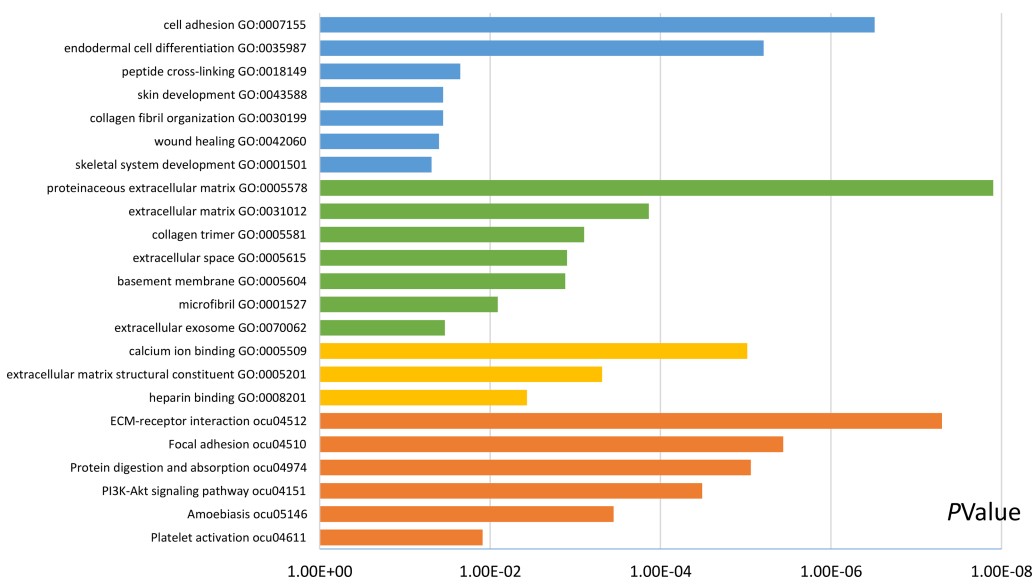

**Figure 4  Results of GO and KEGG pathway analyses of 20 hub genes.** The blue color represents BP, the green color represents CC, the yellow color represents MF, and the orange color represents KEGG pathways.

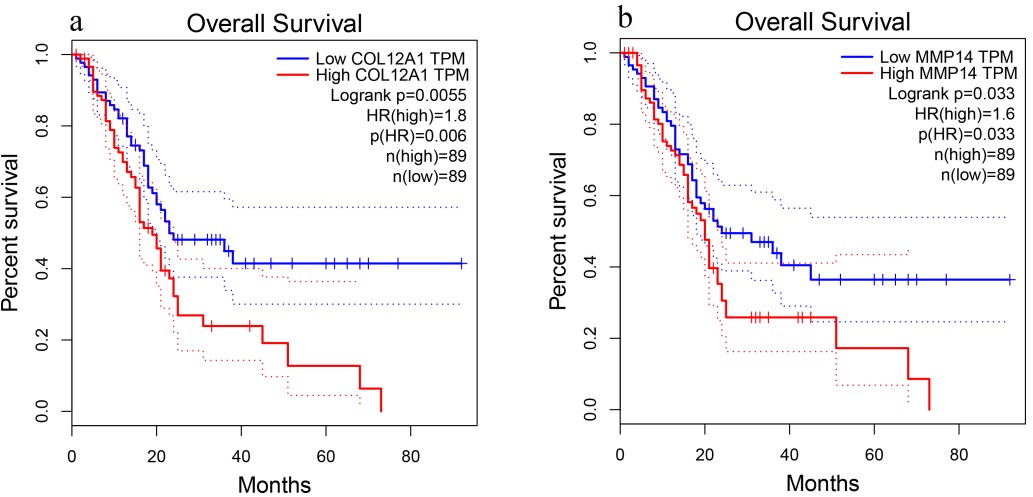

**Figure 5  Overall survival analysis.** Overall survival curves for (A) *COL12A1* and (B) *MMP14* expression in PDAC patients in comparison with a high-risk group and a low-risk group. A value of $p < 0.05$ was regarded as statistically significant. TPM, transcripts per million; HR, hazards ratio.

with an unfavorable prognosis in PDAC patients ($p < 0.05$) (Fig. 5). The overall survival analysis showed that the other hub genes had no statistically significant correlations ($p > 0.05$).

**Table 3 Results of GO and KEGG pathway analyses of 20 hub genes.**

| Pathway ID | Pathway description | Count in gene set | $p$-value | FDR | DEGs |
|---|---|---|---|---|---|
| GO:0007155 | Cell adhesion | 6 | 3.07E−07 | 3.44E−04 | COL12A1, POSTN, VCAN, COL8A1, THBS2, FN1 |
| GO:0035987 | Endodermal cell differentiation | 4 | 6.17E−06 | 0.006927836 | COL12A1, MMP14, COL8A1, FN1 |
| GO:0018149 | Peptide cross-linking | 2 | 0.02230059 | 22.35847168 | COL3A1, FN1 |
| GO:0043588 | Skin development | 2 | 0.035457234 | 33.30840338 | COL3A1, COL5A2 |
| GO:0030199 | Collagen fibril organization | 2 | 0.035457234 | 33.30840338 | COL3A1, COL5A2 |
| GO:0042060 | Wound healing | 2 | 0.039805932 | 36.60570116 | COL3A1, FN1 |
| GO:0001501 | Skeletal system development | 2 | 0.048448486 | 42.72191792 | FBN1, COL5A2 |
| GO:0005578 | Proteinaceous extracellular matrix | 7 | 1.25E−08 | 1.13E−05 | MATN3, BGN, FBN1, COL12A1, POSTN, COL1A1, FN1 |
| GO:0031012 | Extracellular matrix | 4 | 1.37E−04 | 0.123843274 | COL12A1, VCAN, MMP14, COL8A1 |
| GO:0005581 | Collagen trimer | 3 | 7.87E−04 | 0.70897958 | COL12A1, COL1A1, COL8A1 |
| GO:0005615 | Extracellular space | 6 | 0.001248571 | 1.122732142 | ALB, FAP, COL3A1, FBN1, COL12A1, VCAN |
| GO:0005604 | Basement membrane | 3 | 0.001310663 | 1.178271997 | FBN1, COL8A1, THBS2 |
| GO:0001527 | Microfibril | 2 | 0.008113224 | 7.097635089 | LTBP1, FBN1 |
| GO:0070062 | Extracellular exosome | 7 | 0.033863185 | 26.75315613 | BGN, ALB, FBN1, COL12A1, COL8A1, FN1, CDH11 |
| GO:0005509 | Calcium ion binding | 7 | 9.57E−06 | 0.007173793 | MMP14, MATN3, LTBP1, FBN1, VCAN, THBS2, CDH11 |
| GO:0005201 | Extracellular matrix structural constituent | 3 | 4.84E−04 | 0.362368483 | COL3A1, FBN1, COL5A2 |
| GO:0008201 | Heparin binding | 3 | 0.003684116 | 2.728227814 | POSTN, THBS2, FN1 |
| ocu04512 | ECM–receptor interaction | 6 | 5.00E−08 | 4.15E−05 | COL3A1, COL6A3, COL1A1, COL5A2, THBS2, FN1 |
| ocu04510 | Focal adhesion | 6 | 3.64E−06 | 0.003011517 | COL3A1, COL6A3, COL1A1, COL5A2, THBS2, FN1 |
| ocu04974 | Protein digestion and absorption | 5 | 8.69E−06 | 0.007197854 | COL3A1, COL6A3, COL12A1, COL1A1, COL5A2 |
| ocu04151 | PI3K-Akt signaling pathway | 6 | 3.26E−05 | 0.026969277 | COL3A1, COL6A3, COL1A1, COL5A2, THBS2, FN1 |
| ocu05146 | Amoebiasis | 4 | 3.54E−04 | 0.292970736 | COL3A1, COL1A1, COL5A2, FN1 |
| ocu04611 | Platelet activation | 3 | 0.012199259 | 9.669074187 | COL3A1, COL1A1, COL5A2 |

## Validation of DEGs using TCGA and GTEx databases

To ensure the reliability of the identification of the top 20 hub genes, we validated these via the GEPIA using TCGA and GTEx databases. Boxplots of the hub genes associated with PDAC were downloaded from the GEPIA. The results demonstrated that *FN1*, *COL1A1*, *COL3A1*, *BGN*, *POSTN*, *FBN1*, *COL5A2*, *COL12A1*, *THBS2*, *COL6A3*, *VCAN*, *CDH11*, *MMP14*, *LTBP1*, *IGFBP5*, *FAP*, *MATN3*, and *COL8A1* were significantly overexpressed in PDAC tissues in comparison with normal pancreatic tissues, whereas *ALB* was underexpressed in PDAC tissues ($p < 0.05$) (Fig. 6). *CXCL12* was expressed in PDAC tissues, but with no statistically significant difference in expression ($p > 0.05$).

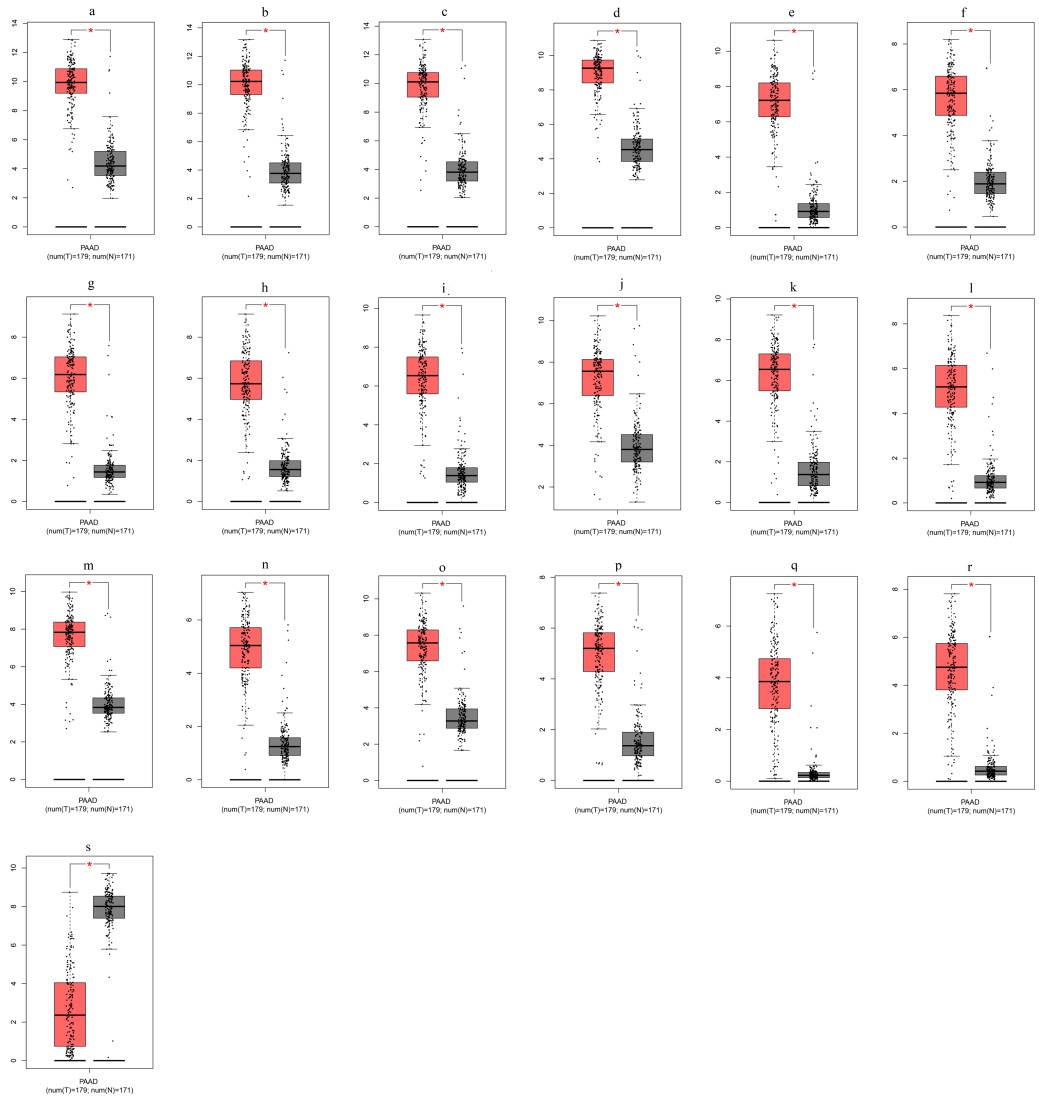

**Figure 6** **Validation of DEGs using The Cancer Genome Atlas and Genotype–Tissue Expression databases.** The boxplots were downloaded from the Gene Expression Profiling Interactive Analysis tool and are arranged in the following order: (A) *FN1*, (B) *COL1A1*, (C) *COL3A1*, (D) *BGN*, (E) *POSTN*, (F) *FBN1*, (G) *COL5A2*, (H) *COL12A1*, (I) *THBS2*, (J) *COL6A3*, (K) *VCAN*, (L) *CDH11*, (M) *MMP14*, (N) *LTBP1*, (O) *IGFBP5*, (P) *FAP*, (Q) *MATN3*, (R) *COL8A1*, and (S) *ALB*. A value of $p < 0.05$ was regarded as statistically significant. The $Y$-axes represent the expression in terms of $\log_2$ (TPM + 1). The red boxes represent the expression levels of DEGs in PAAD tissues, whereas the gray boxes represent the expression levels of DEGs in normal tissues. PAAD, pancreatic adenocarcinoma.

## Validation of expression of candidate gene-encoded proteins

We obtained the expression levels of proteins encoded by the 20 hub genes associated with PDAC from the HPA website. No data for proteins encoded by *COL5A2*, *IGFBP5*, and *MATN3* are reported on the HPA website, and expression profiles of the other 17 genes in PDAC clinical specimens are shown in Fig. 7.

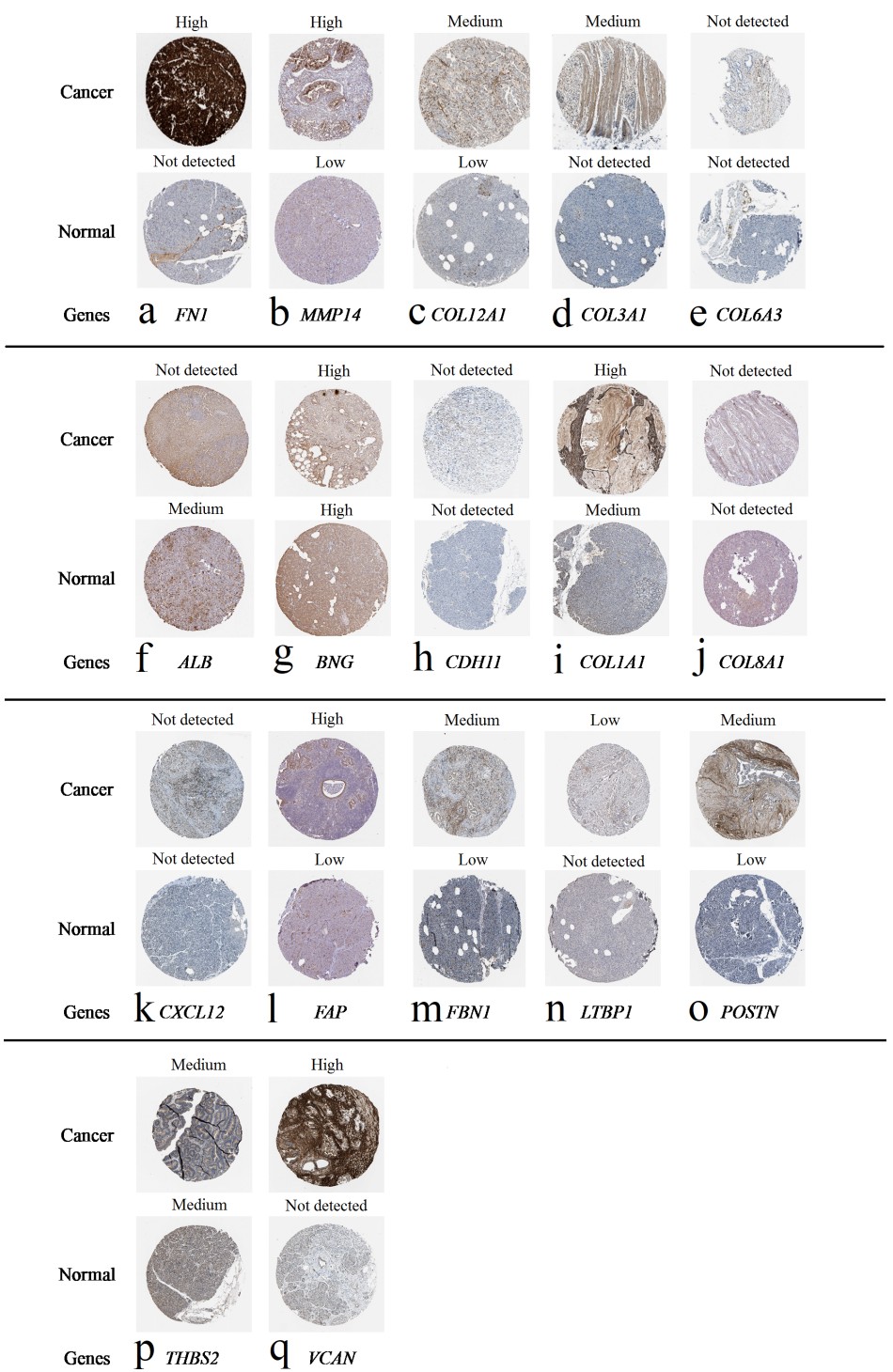

**Figure 7** **Expression of 20 candidate DEGs in human pancreatic cancer specimens.** The immunohisto­chemical data were obtained from the Human Protein Atlas. Except for *COL5A2*, *IGFBP5*, and *MATN3*, expression profiles of the other 17 genes in PDAC clinical specimens are shown. Staining demonstrated (continued on next page...)

**Figure 7 (…continued)**
that the protein expressions of (A) *FN1*, (B) *MMP14*, (C) *COL12A1*, (D) *COL3A1*, (I) *COL1A1*, (L) *FAP*, (M) *FBN1*, (N) *LTBP1*, (O) *POSTN*, and (Q) *VCAN* were higher in PDAC tissues than in normal pancreatic tissues, with only (F) *ALB* being downregulated in PDAC tissues. (E) *COL6A3*, (H) *CDH11*, (J) *COL8A1*, and (K) *CXCL12* were not expressed, whereas (G) *BGN* and (P) *THBS2* were overexpressed in both PDAC tissues and normal tissues.

The protein expressions of *FN1*, *MMP14*, *COL12A1*, *COL3A1*, *COL1A1*, *POSTN*, *VCAN*, *LTBP1*, *FBN1*, and *FAP* were upregulated in PDAC tissues in comparison with normal tissues, with only *ALB* being downregulated in PDAC tissues. *COL6A3*, *COL8A1*, *CDH11*, and *CXCL12* were not expressed in either PDAC tissues or normal tissues, and *BGN* and *THBS2* were overexpressed in both cancer and normal tissues.

## DISCUSSION

Our study was based on GEO datasets, namely, GSE28735, GSE62165, and GSE91035. The main findings deduced from the studies used to compile GSE28735 were that dipeptidase 1 and a unique set of free fatty acids played roles in the development, progression, and prognosis of PC and might be potential targets in PDAC (*Zhang et al., 2012*; *Zhang et al., 2013*). The study that was used to compile GSE62165 found that hepatocyte nuclear factor (HNF)-1 $\alpha$ and HNF-1 $\beta$ seem to be good candidates as tumor suppressors in PDAC (*Janky et al., 2016*). Another paper, which was used to compile GSE91035, concluded that an increase in the expression of the processed transcript of *HNRNPU* was associated with a poor prognosis in PDAC (*Sutaria et al., 2017*).

In our study, GO analysis showed that the most significantly enriched BP, CC, and MF terms among the 20 hub genes were cell adhesion, proteinaceous extracellular matrix, and calcium ion binding, respectively. Cell adhesion is the attachment of a cell either to another cell or to an underlying substrate. The proteinaceous extracellular matrix provides structural support and biochemical or biomechanical cues for cells or tissues and is a structure located external to one or more cells. The ECM is a crucial factor in both promoting the progression of PDAC and inhibiting the delivery of antitumor therapy (*Weniger, Honselmann & Liss, 2018*).

According to the analysis of the MF terms among the hub genes, *MMP14*, *THBS2*, *CDH11*, *FBN1*, *LTBP1*, *MATN3*, and *VCAN* were jointly involved in calcium ion binding, which is defined as selective and non-covalent interactions with calcium ions ($Ca^{2+}$). $Ca^{2+}$ is a ubiquitous and versatile second messenger involved in the regulation of numerous cellular functions, including gene transcription, vesicular trafficking, and cytoskeletal rearrangements (*Nunes-Hasler, Kaba & Demaurex, 2020*). $Ca^{2+}$ and $Ca^{2+}$-regulating proteins contribute to a large number of processes that are key to cancer cells, including proliferation, invasion, and cell death (*Monteith, Prevarskaya & Roberts-Thomson, 2017*; *Prevarskaya, Skryma & Shuba, 2011*). A high serum $Ca^{2+}$ level is associated with a poor prognosis in PDAC (*Dong et al., 2014*), and cytosolic $Ca^{2+}$ overload triggers apoptotic death pathways (*Brini & Carafoli, 2009*). It is thus reasonable that the seven abovementioned genes might regulate calcium ion binding and affect the development of PDAC. Furthermore, our study suggests that *MMP14* is a promising biomarker for survival

in PDAC. Considering that $Ca^{2+}$ cannot be produced in cells but undergoes flux between intracellular calcium storage, cytosolic calcium signals, and the extracellular calcium pool (*Yang et al., 2020*), it would be reasonable to hypothesize that the overexpression of *MMP14* influences calcium ion storage and thus might cause disorders of calcium homeostasis and hence contribute to an unfavorable prognosis in PDAC patients.

Matrix metalloproteinases (MMPs) are a family of calcium- and zinc-dependent membrane-anchored or secreted endopeptidases that are overexpressed in various diseases, including breast cancer (*Min et al., 2015*). MMP14 is located in neoplastic epithelium. It is speculated that the overexpression of *MMP14* alone may be sufficient to induce the development of PDAC (*Shields et al., 2012*). Moreover, *MMP14* is overexpressed in the epithelium in invasive PC (*Iacobuzio-Donahue et al., 2002*; *Shields et al., 2012*), and MMP14, as an endopeptidase, can degrade various components of the ECM such as collagen, which possibly leads to metastasis of tumors (*Golubkov et al., 2010*). Type I collagen can induce the expression of *MMP14* and *TGF-β1* in pancreatic ductal epithelial cells (*Ottaviano et al., 2006*), and *COL1A1* encodes the major component of type I collagen. The expression of *MMP14* in PDAC cells stimulates pancreatic stellate cells (PSCs) and enhances the production of type I collagen by increasing transforming growth factor-β signaling (*Krantz et al., 2011*). Ottaviano et al. found that fibrosis and the expression of *MMP14* in tumor specimens increased in comparison with those in normal pancreatic tissue (*Ottaviano et al., 2006*). These findings suggest the key role of interactions between *MMP14* and type I collagen in the progression of PDAC and support *MMP14* as a potential target for inhibiting fibrosis, preventing metastasis, and treating PDAC.

The KEGG pathway analysis showed that six hub genes, namely, *COL1A1*, *COL3A1*, *COL5A2*, *COL6A3*, *FN1*, and *THBS2*, were significantly associated with ECM–receptor interactions, focal adhesion, and the phosphatidylinositol-3-kinase–protein kinase B (PI3K-Akt) signaling pathway. In addition, collagen-encoding genes, including *COL1A1*, *COL3A1*, and *COL5A2*, were also enriched in protein digestion and absorption and platelet activation.

ECM–receptor interactions play important roles in the processes of cell shedding, adhesion, degradation, migration, differentiation, hyperplasia, and apoptosis (*Bao et al., 2019*). PSCs secrete several ECM proteins, including collagen, fibronectin, fibulin-2, and laminin, as well as hyaluronan (*Hall et al., 2019*). Moreover, *COL1A1* and *COL3A1* were significantly downregulated in PC ($p <0.0001$) after treatment with gemcitabine in combination with EC359 (*Hall et al., 2019*). The gene *COL1A1* encodes the pro-alpha 1 chain of type I collagen, which is closely associated with *MMP14*. *COL3A1* was found to encode a major structural component of hollow organs such as large blood vessels, the uterus and bowel, and tissues that must withstand stretching (*Kuivaniemi & Tromp, 2019*). As an important molecule, *COL5A2* is associated with remodeling of the ECM and is differentially expressed between in situ ductal carcinoma and invasive ductal carcinoma (*Vargas et al., 2012*). The alpha 3 chain of type VI collagen is mainly present in the desmoplastic stroma in PDAC, with large deposits between the sites of stromal fatty infiltration and around the malignant ducts (*Arafat et al., 2011*), and the circulating form of this protein has potential clinical significance in the diagnosis of pancreatic malignancy

(*Kang et al., 2014*). *FN1* encodes a collagen-associated protein that has been identified as a potential biomarker of an unfavorable prognosis in PDAC (*Hu et al., 2018*). *THBS2* appears in the early stages of PDAC and hence has great potential for the diagnosis of PDAC, with 98% specificity (*Kim et al., 2017*).

At points of ECM–cell contact, specialized structures are formed, which are termed focal adhesions. Some components of focal adhesions contribute to cell migration in PDAC and participate in structural links between the actin cytoskeleton and membrane receptors, whereas others are signaling molecules (*Manoli et al., 2019*).

The PI3K-Akt signaling pathway regulates fundamental cellular functions, including transcription, translation, proliferation, growth, and survival. Accumulating evidence has implied that the PI3K-Akt signaling pathway promotes malignant processes of PDAC cells, including proliferation, angiogenesis, metastasis, suppression of apoptosis, and chemoresistance, and targeting the PI3K-Akt signaling pathway has been a potential therapeutic strategy for the treatment of PC (*Ebrahimi et al., 2017*).

In PC, both exocrine and endocrine functions are abnormal, which profoundly influences the secretion of proteases, and hence protein digestion and absorption is a prominent metabolic change (*Gilliland et al., 2017*). Platelet activation facilitates the P-selectin- and integrin-dependent accumulation of cancer cell microparticles and promotes tumor growth and metastasis (*Mezouar et al., 2015*). However, the effect of collagen-mediated platelet activation on the progression of PDAC needs further investigation.

Collagens are centrally involved in the formation of fibrillar and microfibrillar networks of the ECM and basement membranes, as well as other structures of the ECM (*Gelse, Poschl & Aigner, 2003*). We further found that the collagen family is closely associated with PDAC. Interestingly, Wang and Li also found that the collagen family and *FN1* have an influence on PC via data mining using a different gene set (GSE15471) (*Wang & Li, 2015*). As we have done, they suggested that *FN1*, together with *COL1A1*, *COL3A1*, and *COL5A2*, may be key molecules in the development and progression of PDAC owing to their involvement in ECM–receptor interactions and focal adhesion pathways. These DEGs were also identified in our study. Furthermore, we found that *COL12A1* and *COL6A3* are probably also key DEGs that influence PDAC, which differs from the results of Wang and Li. Although the specific relationship between *COL12A1* and PDAC has not been reported, our findings also suggest that *COL12A1* is a potential prognostic biomarker in patients with PDAC.

We also found that *FBN1* and *COL8A1* appear to be involved in the progression of PDAC. *FBN1* encodes a structural component of the microfibrils of the ECM that have diameters of 10–12 nm, which impart both regulatory and structural properties to load-bearing connective tissues (*Lee et al., 2004*). The silencing of *FBN1* inhibits the proliferative, migratory, and invasive activities of gastric cancer cells, whereas upregulation of the expression of *FBN1* has the opposite effect (*Yang, Zhao & Chen, 2017*). *COL8A1* encodes a macromolecular component of the subendothelium (*Xu et al., 2001*). It is suggested that *COL8A1* may be associated with malignant processes of hepatocarcinoma (*Zhao et al., 2009*) and the progression and prognosis of human colon adenocarcinoma (*Shang et al., 2018*).

## CONCLUSIONS

In conclusion, we screened the top 20 hub genes (*FN1*, *COL1A1*, *COL3A1*, *BGN*, *POSTN*, *FBN1*, *COL5A2*, *COL12A1*, *THBS2*, *COL6A3*, *VCAN*, *CDH11*, *MMP14*, *LTBP1*, *IGFBP5*, *ALB*, *CXCL12*, *FAP*, *MATN3*, and *COL8A1*) and the related enriched functions or pathways, which regulate the progression and metastatic invasion of PDAC, as well as overall survival. The results demonstrate that the upregulation of *MMP14* and *COL12A1* in PDAC is closely associated with poor overall survival, that these might be a potential combination of prognostic biomarkers in patients with PDAC, and that *FBN1* and *COL8A1* might be biomarkers of PDAC. In brief, our study increases the understanding of the potential critical genes and related pathways that participate in the pathogenesis of PDAC.

### Funding

The research was supported by the Special Program of Science and Technology Research of Chinese Medicine Administration Bureau of Sichuan Province (2018JC012). The funders had no role in study design, data collection and analysis, decision to publish, or preparation of the manuscript.

### Grant Disclosures

The following grant information was disclosed by the authors:
Special Program of Science and Technology Research of Chinese Medicine Administration Bureau of Sichuan Province: 2018JC012.

### Competing Interests

The authors declare there are no competing interests.

### Author Contributions

- Jingyi Ding conceived and designed the experiments, performed the experiments, analyzed the data, prepared figures and/or tables, authored or reviewed drafts of the paper, and approved the final draft.
- Yanxi Liu performed the experiments, authored or reviewed drafts of the paper, and approved the final draft.
- Yu Lai conceived and designed the experiments, authored or reviewed drafts of the paper, and approved the final draft.

### Data Availability

Three microarray datasets are available at NCBI GEO: GSE28735, GSE62165, GSE91035. All the original figures and tables are available as Supplemental Files.

### Supplemental Information

Supplemental information for this article can be found online at http://dx.doi.org/10.7717/peerj.10419#supplemental-information.

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
