# Peer review of "Identifying MMP14 and COL12A1 as a potential combination of prognostic biomarkers in pancreatic ductal adenocarcinoma using integrated bioinformatics analysis"

_PeerJ, doi:10.7717/peerj.10419_

## Round 0.1 · original submission · Major Revisions

All reviewers raised a number of concerns about your study. Please address all the critiques and revise your manuscript accordingly.

Reviewer 1 ·

Basic reporting

The manuscript meets basic reporting criteria. The current literature in the field is nicely described with relevant citations. The research question is clearly outlined.

Experimental design

Authors use previously published data and perform a comprehensive bioinformatics analysis to study the differential expression in genes under PC vs normal tissue. All the methods are well described with enough details for the data reproducibility and transparency.

Validity of the findings

The findings are interesting and may help understand the biology of PC in more details, but certainly lack biochemical validation by authors.

Additional comments

Manuscript by Ding and colleagues, presents a bioinformatics study using the previously published microarray datasets on Pancreatic ductal adenocarcinoma (PDAC). Authors identify several differentially expressed genes (DEGs) followed by attempts to understand their biological functions and relevance. The authors further show that the expression of two DEGs viz. COL12A1 and MMP14 was correlated with a poor prognosis in PDAC and poor overall survival; presenting these two DEGs as potential biomarkers for PDAC.
1) The manuscript presents some interesting findings, but it certainly lacks biochemical validation of the differential expression of the genes identified. It mostly represents a consolidation of the previously published dataset and previously published literature (data mining). The manuscript would gain much if authors validate these findings by some experimental evidence like RT-qPCR, immunoblot analyses. Authors may also take a few candidate genes and show any mechanistic data that these are somehow regulated/regulating in PC development. For example – lines 196-207 and lines 208-223 describe speculative mechanistic implications of the findings. Can authors test some of these ideas?
2) Authors should explain the rationale of using the GEO dataset viz. GSE28735, GSE62165, and GSE91035. Why did they select these datasets? Also, it would be very relevant to discuss what were the main findings from the papers that generated these datasets.
3) Figure 1: Are these overlaps statistically significant?
4) Figure 7: Can authors quantify the signals?
5) What are the Y-axes in Figure 6?

Reviewer 2 ·

Basic reporting

Ding et.al. identified 20 differentially expressed genes in Pancreatic ductal adenocarcinoma patients, wherein, MMP14 and COL12A1 is associated with poor overall survival, and these might be a combination of prognostic biomarkers in PDAC. In addition, FBN1 and COL8A1 are probably novel biomarkers of PDAC. Overall, the authors listed a lot of work in the text, I would suggest the author to purify their result and clearly show the most interesting ones.

Experimental design

no comment

Validity of the findings

no comment

Additional comments

no comment

Reviewer 3 ·

Basic reporting

The authors conducted extensive bioinformatics analysis to identify key genes and cellular pathways that play a role in development and progression of pancreatic ductal adenocarcinoma. The authors analyzed 3 different microarray datasets to identify differentially expressed genes. They further used STRING to establish protein-protein interaction networks and performed GO analysis and KEGG pathway enrichment analysis to identify the most significantly enriched biological processes, molecular functions, cellular components, and cellular pathways in PDAC. They validated their findings using the TCGA and GTex databases as well as the expressed proteins using the HPA website. They also investigated the impact of the expression levels of their top 20 hub genes on the overall survival. Considering the notorious lethality of PDAC, it is important to perform bioinformatic studies such as the work in this manuscript to contribute to the current efforts to identify prognostic biomarkers. This work is within the scope of PeerJ journal. The manuscript is written in professional English and the structure conforms to PeerJ standard. Most figures are of publication quality and appropriately annotated. Most of the raw data relevant to the manuscript is available. A few points are noted below which must be addressed prior to acceptance.
1. Figure 2C: Please add a color-key explaining what the different shades indicate in terms of the scores for each hub gene.
2. Figure 6: The individual images are of good quality, however, the ones in the PDF are not clear. The images in the PDF need to be improved in terms of their readability. Also, please describe what the colors (red and gray) indicate.
3. Section 3.4: Please provide overall survival data for other hub genes as well.

Experimental design

The authors have defined the objective of the study well. The bioinformatic analyses are well structured and seem to be performed accurately following standard procedures in the field. However, the authors must clarify the following points to improve the overall clarity and transparency of the manuscript.
1. Section 2.1: The authors have used three data sets- GSE28735, GSE62165 and GSE91035 while there are several other datasets related to pancreatic cancer available in the GEO database. The authors must provide the rationale for choosing these particular datasets.
2. Section 2.2: Please explain the method used to obtain the adjusted p-values
3. Section 2.3: Please provide the version number of the cytoscape software used

Validity of the findings

Overall, the results are detailed and presented in a clear manner for the most part. Appropriate statistical analyses have been performed and reported in accordance with standards in the field. The discussion of results is thorough, and the conclusion answers the original research question. There are some points outlined below which must be rectified/ clarified prior to acceptance of the manuscript.
1. Line 149: “Enrichment of KEGG….” This sentence is not required here and fits better in the next section where the authors have already explained this in lines 161-163.
2. Section 3.3: Is there a set cut-off based on which certain functions or pathways are designated as “enriched”? Based on figure 3 & 4, it appears that to call a certain function or pathway “enriched”, the p value needs to be less than 0.0001. The authors should clearly state what the criteria is. If it is the case that anything with p<0.0001 is considered “enriched”, then why is ‘endodermal cell differentiation’ not considered in the hub genes?
3. Section 3.4: Provide overall survival data for other hub genes as well. Line 168 should say “…demonstrated that the high expression …”
4. Section 3.5: CXCL12 is missing from this list of genes. Is there any specific reason for this or is this an accidental omission? If it was accidentally omitted, please, add a panel for this gene in figure 6 and comment on the expression trend.
5. Section 3.6: Certain panels in figure 7 do not support the conclusions drawn. For FBN1, both normal and cancer panels look identical. I believe, the same image might have been pasted in both cases. The separate images supplied as supplemental files with the manuscript looks correct. Please correct the ‘normal’ panel for FBN1. Based on the images, CDH1 looks to be overexpressed in cancer tissue whereas the results state that this protein was not expressed in PDAC or normal tissues. Please clarify this ambiguity. Similarly, ALB appears to be upregulated in cancer tissue compared to the normal tissue. Please clarify this ambiguity.
6. For tables 1 & 2: Add another column listing the different genes involved in each BP, CC and MF terms as well as pathways. This makes it easier to tie it back to the discussion section.
7. Line 205-207: The basis for this statement is not clear. The authors should elaborate how they concluded that interaction of MMP14 with Ca2+ can serve as a biomarker. Please provide overall survival data supporting this claim.
8. Line 246: please cite the full reference.
9. Line 291: “FBN1 and COL8A1 might be biomarkers of PDAC”. The basis of this statement is not clear. How do the authors explain this conclusion given that in section 3.6 they mention that the protein product of COL8A1 was not expressed in cancer or normal tissue? Please provide a more detailed justification for this statement.

---

## Round 0.2 · accepted · Accept

Both reviewers pointed out that all their critiques were addressed and the manuscript was amended accordingly. Therefore, I am pleased to accept the revised version of your manuscript.

Reviewer 1 ·

Basic reporting

The manuscript meets the basic criteria of reporting.

Experimental design

The manuscript employs sound experimental design.

Validity of the findings

The findings are valid and supported by the experimental results.

Additional comments

In this revised version, the authors have addressed the concerns raised by the reviewers. I recommend this article for publication in PeerJ.

Reviewer 3 ·

Basic reporting

The revised manuscript and the rebuttal letter address the previously raised concerns appropriately.

Experimental design

The revised manuscript and the rebuttal letter address the previously raised concerns appropriately.

Validity of the findings

The revised manuscript and the rebuttal letter address the previously raised concerns appropriately.